# Systematic Literature Reviews in Kansei Engineering for Product Design—A Comparative Study from 1995 to 2020 [note 1]

**DOI:** 10.3390/s21196532

**Published:** 2021-09-30

**Authors:** Óscar López, Clara Murillo, Alfonso González

**Affiliations:** 1Department of Mechanical, Energy and Materials Engineering, University Centre of Merida, 06800 Mérida, Spain; agg@unex.es; 2IT and Telematics Services Engineering, School of Technology of Cáceres, 10003 Cáceres, Spain; claramr@unex.es

**Keywords:** product design, Kansei engineering, systematic literature review, comparative study

## Abstract

Individual products and models on the market must be specifically differentiated from the rest to meet user demand. In terms of consumer purchasing behaviour, consumers increasingly base their decisions on subjective terms or the impression that the product leaves on them, both in terms of functionality, usability, safety, and price adequacy, and regarding the emotions and feelings that it triggers in them. This demand has lead both Asia and Europe to implement new methodologies to develop new products, such as “emotional design” or Kansei engineering. This paper presents a systematic literature review (SLR) on the most relevant methodologies based on Kansei engineering and their relevant results in the specific discipline of product design, addressing these five questions: (RQ1) How many studies on KE and emotional design are there in the Scopus and Web of Science (WoS) databases from 1995 to February 2021? (RQ2) Which research topics and types of KE are addressed? (RQ3) Who is leading the research on KE and emotional design? (RQ4) What are the benefits and drawbacks of using and applying the methodology? (RQ5) What are the limitations of the current research? We analysed 87 studies focusing on the Kansei methodology used for product design and device technologies (e.g., shape design, actuators, sensors, structure) and aesthetic aspects (e.g., Kansei words selection, the quantification of measured emotions of results, and detected shortcomings), and provided the database with all the collected information. One identified and highlighted sector in the results is the electronic–technological-device sector. Results confirm that this type of methodology has a majority and direct application in these sectors, and they are widely represented in the automotive and electronics industries. Lastly, this SLR provides researchers with a guide for comparative emotional-design work, and facilitates future designers who want to implement emotional design in their work by selecting the specific type according to the results of the SLR.

## 1. Introduction

These days, product design must address areas such as competition [1], marketing [2], and processes [3,4]. Customers increasingly buy on the basis of subjective terms and the impression that they have of a product [5]. In today’s market, consumers value the functionality, usability, safety, and price appropriateness of products, and the emotions and feelings that they trigger. In an increasingly competitive market, a good product should meet all consumer expectations, but especially provoke a positive emotional response.

Therefore, manufacturers need an instrument to predict how an item is received in the market. This demand has lead both Asia and Europe to pursue a new field of research on the translation of these subjective customer impressions into concrete products. This field is known as “emotional design” [6].

The product itself often evokes emotions in consumers with different dimensional and interactive design variables, and this is performed through various methodologies. Affective engineering or Kansei, developed in the early 1970s at Hiroshima University [7,8,9], aims to improve products or services by translating people’s feelings or psychological needs into product parameters. It is a methodology within affective engineering [10].

Products can provoke multiple simultaneous emotions in the user, some of which may even be contradictory. This, combined with the personal differences that each designer can bring to the development of their professional activity [11], makes it necessary to provide guidance on the best typology or the one most used by other professionals with more experience, if this helps to develop the work with a higher guarantee of success.

Moreover, there are still many regions in the world where, despite the confirmation that KE is one of the best methodologies for designing products from an emotional viewpoint, there are still difficulties in applying it [12].

“Kansei” refers to a concept in psychology regarding the integration of consumers’ different senses (vision, hearing, smell, touch, etc.) and cognition caused by the size, colour, performance, price, and other factors of the product, i.e., product attributes [13]. This can be applied to different areas of design, such as physical product design [14], web interface design [15], or services [16,17].

The basic principles of this method are the identification of product properties and the correlation between these properties and design features. This method has three pivotal points: how to accurately understand the Kansei consumer, how to reflect and translate the understanding of Kansei into product design, and how to create a system and organisation for Kansei-orientated design [18].

Several types of Kansei engineering were identified in various contexts. Nagamachi [19] collected all of the applications of Kansei engineering that he had produced and grouped them according to the used tools and performed task areas. From these groups, he identified what are known as Kansei engineering types. To date, eight types of KE were classified by several researchers [9,20,21,22]. Figure 1 shows the Kansei engineering (KE) framework that Lokman [23] developed to summarise the principles of applying KE. It comprises the eight types of KE classified by various researchers [9,20,21,22,23]. This list may be expanded at any point in time, as they are constantly evolving.

Simon Schütte examined these types of Kansei engineering and developed a general model that covers the most relevant contents [24].

Although emotional design promises results, evidence is needed of where it can be applied, in which areas it offers the best results, which characteristics have the most promising methodologies, and what improvements can be made to the product, always considering how to improve the user experience (UX). Each user interacts particularly with products, and the results of these interactions are captured in the user’s emotions and expectations [25]. This user experience can be identified through the emotional responses that the user experiences when using a product, and represents an evolution in users’ perception of usability [26].

To the authors’ knowledge, there is only one previous study that provides a basic and non-in-depth analysis of emotional-design methodologies, where they have been applied, what improvements they have brought, whether they have been tested, and on what they are based [27]. Therefore, this paper presents an indepth systematic literature review (SLR) with an analysis and discussion of new data on the most relevant methodologies based on emotional design and their proven results in the specific discipline of product design.

The availability of this SLR provides researchers with a guide for comparative emotional design work and facilitate future designers who want to implement emotional design in their work.

The rest of this paper is structured as follows: Section 2 explains the background to the research; Section 3 presents the methodology for systematically reviewing the state of the art of Kansei engineering research and emotional design; Section 4 examines and analyses the findings, and discusses the significance of the results for the academic and professional world; lastly, Section 5 presents the conclusions of this systematic review.

## 2. Background

Emotional design is a design approach focusing on creating products that offer positive user experiences [28]. The importance of emotions on the human capacity to understand the world and the learning process, including information processing and decision making, is raised.

To better understand these processes, a hierarchy of human needs is necessary [29], which is applied to human factors to propose an order of consumer needs consisting of functionality, usability, and pleasure [30].

In recent years, much progress has been made on the basis of this proposal, and emotions are increasingly important in the field of design and engineering [31]. Thus, new design methods were developed to generate user sensations beyond those derived from the product and its functionality, such as associating certain sensations provoked by competition or associating certain sensations triggered by products with their respective brand, leading to improvement in competition [32].

Therefore, there is a clear link with the user experience concept, of which the main objective is to achieve an affective connection between the product or brand and the user or consumer. An understanding of this strategy must involve analysing emotional design. In this sense, specific authors such as Arhippainen and Tähti defined UX as the user’s sensations when interacting with a product [25].

Before this statement, Dillon proposed a model that defined UX as a sum of three levels: Action, what the user does; Result, what the user obtains; and Emotion, what the user feels. Much more importance is thus given to the emotions that it provokes in the user [33].

Emotional design can be applied to many areas. The most abundant research works on this topic deal with human–machine interaction (HMI), as the acceptance of the product by the user often depends on generated sensations [34].

Applying emotional design in technological products or household appliances is widespread. In these areas, competition is very high, and purchasing one of these products can be a significant investment for the customer, so the emotion that it triggers can be the reason for the decision [35].

There are also several research papers linking emotional design to areas of healthcare, such as physiotherapy [36], orthopaedics [37] or elderly care [38]. Emotions can be critical in patients’ recovery.

There are emotional design methodologies based on Kansei engineering with different adaptations, such as applying the Rasch model [39] or big data [40], building QFD matrices [41] or making use of aesthetic intelligence [42] and some work reviewing design methodologies in general [43]. However, no review compiles relevant methodologies based on emotional design and compares them.

## 3. Methodology

The research is based on a structured literature review from 1995 to February 2021, grounded in a systematic, method-based, and replicable approach [44,45,46,47].

An SLR searches, assesses, synthesises, and analyses all studies relevant to a specific research field [48,49]. According to Tranfield [47], an SLR is characterised as a scientific and transparent process that minimises bias through comprehensive literature searches and by providing an audit trail of the reviewer’s procedures. There are different studies on how to perform an SLR in different fields, such as Kitchenham [50,51] in software engineering, Tranfield [47] and Nightingale [52] in health, and vom Brocke et al. [53] in Industry 4.0. An SLR was conducted to evaluate current methodologies based on emotional design and identify the principal methodologies that apply Kansei engineering to a product. In this paper, the SLR approach suggested by B. Kitchenham and Charters [50] is applied. Besides achieving the above objectives, the SLR verifies success stories in product applications. An SLR is a type of scientific research that objectively and systematically integrates the results of empirical studies on a given research problem. On the basis of our baseline study [50], five consecutive steps are defined as follows:Define research questions. The main objective of this SLR summarises methodologies based on emotional design applied to products, identifying the amount and type of research, and the available research results. On the basis of specific research questions, the aim was to arrive at the documents that best suited the purpose of this study.Conduct a literature search. Primary studies are identified by using search strings in scientific databases. An excellent way to create a search string is to structure it around collection, intervention, comparison, and result. Structures are based on the research questions.Selection of studies. Inclusion and exclusion criteria are used to exclude studies that are not relevant to answer the research questions. For example, while some papers applied an emotional design methodology to a product, they exclusively focused on physical parameters such as product measurements. A systematic three-stage process was followed for this selection:(a)The title, abstract, and keywords of each document were analysed to decide whether to discard them or not according to the inclusion and exclusion criteria, which are defined in more detail in Section 3.3.(b)The same elements of each document were evaluated to classify them according to the defined questions in this text.(c)All selected papers were carefully analysed to refine the assessment.Quality assurance. The journals of the selected studies were verified to see whether they were indexed in the Journal Citation Report and in which quartile, and in other rankings such as SCImago Journal Rank and Scopus.Data extraction and management. After analysing the selected documents, each article’s common and differential characteristics were extracted and assessed in a table.

Each of the SLR steps was then applied to the research question being addressed.

### 3.1. Research Questions

Although the general objective of this study could be summarised in the analysis of the most relevant methodologies based on emotional design applied to a product case, this objective is explained in five specific research questions to gain a more detailed understanding of the topic, as follows.

RQ1. How many studies on KE and emotional design are there in the Scopus and Web of Science (WoS) databases from 1995 to February 2021?RQ2. Which research topics and types of KE are addressed?RQ3. Who is leading the research on Kansei engineering and emotional design?RQ4. What are the benefits and drawbacks of using and applying the methodology?RQ5. What are the limitations of the current research?

The primary purpose of these research questions is to analyse the number of studies on emotional design and KE methodologies in a specific period. The secondary aim is to recognise the strengths and limitations of applying these methodologies in this field of research.

To address RQ1, an extensive time period must be identified to help interpret how the application of the methodologies has evolved. Regarding RQ2, the specific topics, typologies, and key aspects that differentiate them were considered. In RQ3, individual researchers and the origin of their research that can be correlated with the application sector were identified. Regarding RQ4, the importance for this study of papers with practical conclusions and a direct application in developing the research itself are highlighted, as they contribute towards identifying the success of the selected typology in terms of the product. Lastly, regarding RQ5, we visualise the possible individual disadvantages of each KE typology in its application, comparing it with the developed or analysed product.

To answer these questions, the authors searched electronic database platforms SCOPUS and WoS—the largest citation databases of the peer-reviewed literature—from their inception until February 2021. These libraries have a broad coverage of publications in science and technology, engineering, and computer science, and index several publication catalogues (including the MDPI, IEEE, ACM, Springer, and Elsevier libraries).

### 3.2. Data Sources and Search Strategy

The search strategy was developed on the basis of the research questions and considering the keywords for each, including synonyms, to refine the search. The search string is shown below:


    (‘‘Kansei Engineering’’) OR (‘‘emotional design’’)



    AND



    (‘‘use’’) OR (‘‘application’’) OR (‘‘used’’)



    AND



    (‘‘results’’) OR (‘‘effect’’) OR (‘‘success’’)



    AND



    (‘‘product’’) OR (‘‘Type Kansei’’)



    AND



    (LIMIT-TO ( DOCTYPE,‘‘ar’’) )



    AND



    ( LIMIT-TO ( EXACTKEYWORD,‘‘Kansei Engineering’’ ) OR



    LIMIT-TO (EXACTKEYWORD,‘‘Product Design’’ ) )



    AND



    ( LIMIT-TO ( LANGUAGE,‘‘English’’ ) )


The selected search engine was the Scopus database. Publications appearing in this library had undergone a peer-review process and were of acceptable quality. The selected journals and conferences are shown in Table 1.

### 3.3. Study Selection

For the first phase of study selection, two simple criteria were defined.

The included studies must meet the following conditions:(a)fall within the time frame of 1995–2021;(b)understand the emotional response of the user in the performance of the work;(c)incorporate the use of technology;(d)be complete articles;(e)belong to the branch of industrial design; and(f)be studies with a design methodology or process based on KE and apply it to a product, test, or prototype.The articles to be excluded must meet at least one of the following conditions:(a)not dealing with emotional design or KE;(b)not explaining and defining a design methodology or design process;(c)published in a language other than English;(d)not indexed in at least two bibliographic references and citation databases;(e)conference paper; and(f)duplicate study.

These selection criteria were dictated by the research questions described in Section 3.1 above. These issues define the main dimensions that need to be considered to implement and deploy a valid solution in the environment described above. The search string resulted in 109 publications. The full text of 87 publications was assessed for eligibility. The identification, screening, and eligibility checking of the studies were performed by the same author (i.e., Clara Murillo).

As explained, a systematic three-stage process was followed for this selection:Phase 1. Only the title, abstract, and keywords of each paper were analysed to establish whether to discard them or not according to the inclusion and exclusion criteria. The number of results in the search string was 109 documents.Phase 2. The same elements of each document were assessed to classify them according to the issues defined in this document. After this second assessment, 22 papers were excluded, as they had met one of the exclusion requirements, namely, they were not accessible at the time of reading, were not indexed according to the requirements mentioned in this SLR, or were not available in English.Phase 3. All papers were carefully analysed to refine the assessment. The total number of carefully reviewed documents was 87.

Figure 2 shows the flow diagram that was used to organise the SLR.

#### Quality Assurance

To assess the quality of the obtained results, the journals to which the studies belonged were analysed to see whether they were indexed in the Journal Citation Report (JCR), SJR, and Scopus. Table 2 shows an outline with the name of the journals, whether or not they were indexed in the JCR, SJR and Scopus, in which quartile, and the impact factor, all in the category of industrial engineering.

### 3.4. Quality Assessment

The SCOPUS and WoS electronic database platforms were used to assess each SLR. The used quality criteria were based on five quality assessment questions (QAQs):QAQ1. Do they offer keywords to assess the emotions that the product triggers in the customer?QAQ2. Does the process incorporate the use of technology and/or artificial intelligence to automate it?QAQ3. Were quantitative scales identified for assessment?QAQ4. Have emotions been considered in product development?QAQ5. Is there a direct application of the methodology to test the obtained results?

The evaluation system was as follows.

QAQ1: Y (Yes), there are at least 10 keywords; P (Partial), there are fewer than 10 keywords; N (No), there are no keywords.QAQ2: Y (Yes), the author incorporates technology and/or artificial intelligence for process automation; N (No), the process is not automated.QAQ3: Y (Yes), quantitative scales are included; P (Partial), a rating scale was included but was not quantitative; N (No), no rating scales are provided.QAQ4: Y (Yes), the author differentiates between emotions and categories; P (Partial), the author only differentiates between emotions or categories; N (No), there is no differentiation by categories and emotions.QAQ5: Y (Yes), there is a direct application of the methodology; P (Partial), the author applies the methodology but does not check it against the results; N (No), the methodology is not applied.

### 3.5. Data Extraction and Management

All selected papers were analysed for this research. This document focuses only on explaining the extracted results to facilitate their understanding (see Table 3).

## 4. Results and Discussion

In searching the highest study, quality we ensured that as many of the analysed articles as possible were indexed and had the highest possible impact index. In this case, we obtained a percentage of over 60% (Figure 3) in the articles analysed and indexed in JCR, and higher in the SCOPUS and SJR databases (Figure 4 and Figure 5).

The set of analysed papers in this study yielded a large amount of data that could be interpreted on many fronts, yielding interesting conclusions from the viewpoint of the typology of KE, the evolution over time, or the importance of carrying out a final phase of application on the product of the obtained data in the first stages of analysis.

Coincidences and similarities or differences between the articles that obtain a high or maximal score according to our scoring system, explained below, are defined through quality assessment questions (QAQs). These QAQs were also weighted according to the importance proposed by the authors themselves (Table 4).

Of the 109 articles obtained from the initial search, 22 were discarded as they were not available for reading and analysis, leaving 87 articles for study and analysis. These were scored in these ranges (Table 5).

The papers that obtained a score equal to or higher than 0.75 out of 1.00, and those that obtained the maximal score of 1.00, were highlighted. In total, 57 articles (66%) were within this range, and 16 articles (18%) achieved the maximal score.

Two of the five quality assessment questions were highlighted by weighting over the total, namely, QAQ4, which highlights the differentiation between emotions and product categories as positive; and QAQ2, which incorporates technology in the analysis of the results.

QAQ2 and QAQ4 reveal that, although there are many adjectives at the beginning of the methodology (QAQ1), the differentiation of these emotions and product categories (QAQ4) and applying technology for the analysis of the results (QAQ2) are necessary to effectively and efficiently develop the process.

QAQ5 was positive in each paper that obtained the maximal score. This allowed for us to interpret that the KE methodology substantially improves if the process directly ends by applying the methodology on the product to improve or carry out the necessary redesign.

Moreover, there is a clear trend towards the KE typology used in the papers that yielded the highest scores, as seen in Table 6 and Table 7, and Figure 6 and Figure 7.

If this analysis is extended to all viewed papers, it shows which methodology the authors prefer (see Figure 8). Although with less incidence, it coincides with the predominant typology in the papers classified with a high score, Type III with 24% and Type VIII with 19%.

Lastly, covered products by each analysed paper were classified into product categories. Figure 9 shows that the areas where KE is most applied are electronics and technology products, and construction and furniture, both urban and household.

After knowing the most relevant product categories that apply Kansei engineering, the most widely used type of KE was analysed.

Figure 10 shows that KE modelling is mostly applied to technological and electronic products. For products related to construction and household products, the most applied typology is rough set, although KE modelling also occupies a high percentage in the type of used KE (Figure 11).

## 5. Conclusions

Given these results, according to the analysed papers in this study, the KE modelling or rough-set typology is preferential. Neither methodology is restrictive in terms of the type of analysed product.

After analysing the product categories that appear in the analysed works, it is concluded that KE is most frequently applied in technology and electronics products such as smart devices (29%) or household appliances and products related to construction and furniture (29%). Again, KE modelling (22%) and Rough Set (26%) appear as the preferred when applying KE. We can conclude that it would be a guarantee of success to use the above mentioned KE typologies when the project is developed in or for technology sectors, home electronics and furniture, and building products.

If we deepen the analysis considering the analysed time horizon, from 1997 to 2021, there is a certain tendency towards the generalised use of these two typologies that gave us such satisfactory results in our analysis of the articles that had obtained high scores (Figure 12). Specifically, KE Modelling is the one most used by the authors in the first years of our study, giving way to the absolute dominance of the Rough Set methodology from 2013 onwards.

As developed by M. Nagamachi in his paper “Kansei Engineering and Rough Sets Model”, it is essential to choose how to measure KE, considering there are emotions that are not linear in their growth and, therefore, it would not be correct to apply the normal distribution with linear growth. The Rough Set methodology solves this incidence by analysing nonlinear or ambiguous data by searching for upper and lower approximations [54].

We conclude that it is critical for users of these methodologies to understand from the outset the benefits of incorporating them into their work processes, which would enable them to readily select the typology that best suits them for their product development depending on the sectors they are in.

These types of methodologies are complex to apply and require guidelines or aids for their proper application. According to Reiser [55], instructional designers often use systematic instructional design procedures and employ various instructional media or technology to accomplish their goals.

Incorporating KE methodologies or typologies into the design in the early stages of learning highlights the designer’s need for guidance or experience or systematic literature reviews to help emphasise the role of emotions for better results [56].

The design sector incorporates an increasing number of technological aspects (products packed with sensors, devices, electronic components and circuits, etc.). One of the identified and highlighted sectors in the results was the electronic–technological-device sector, which confirmed that this type of methodology has a majority and direct application in these sectors, and is widely represented in the automotive and electronics industry, besides many others. In this sense, there are countless applications integrating sensors and communication technologies where they are increasingly used, including robotics, domotics, testing and measurement, do-it-yourself (DIY) projects, Internet of Things (IoT) devices in the home or workplace, science, technology, engineering, education, and the academic world for science, technology, engineering, and mathematics (STEM) skills.

This paper supports design professionals in defining the elements and components of these products, helping to establish a direct correspondence between the users’ emotional response and the component.

Relevant techniques were used to analyse the methodology based on, for example, multivariate analysis and artificial intelligence.

### Research Limitations

This research has its limitations. The next step is to specify and analyse other, more specific conditioning factors.

This study focused on five quality issues presented by the authors. However, other considerations were left out, such as product attributes and characteristics or other differential and significant aspects and determinants when designing products, such as price, customer purchasing power, or environmental sustainability [57].

The authors understand the need for further research into the different KE methodologies and their direct application to the specific expressed conditioning factors, so they can also be compared with the analytical tools used by the KE typologies themselves.

## Figures and Tables

**Figure 1 sensors-21-06532-f001:**
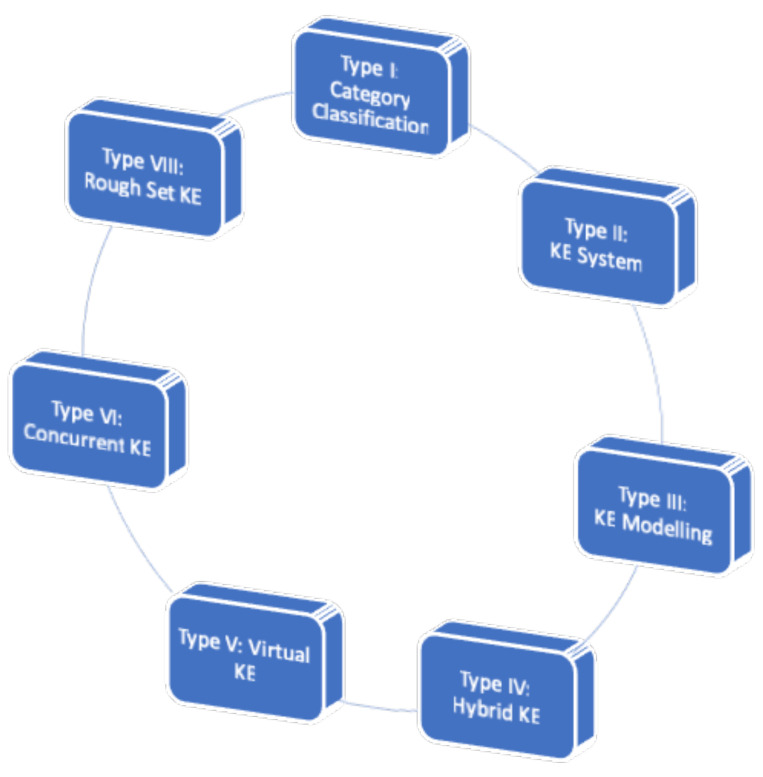
Types of Kansei engineering [23].

**Figure 2 sensors-21-06532-f002:**
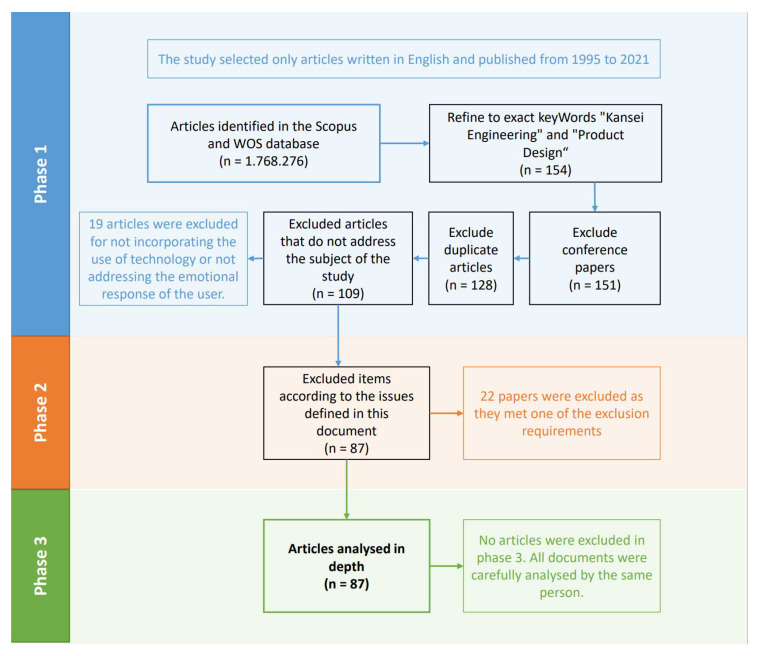
SLR process flow diagram.

**Figure 3 sensors-21-06532-f003:**
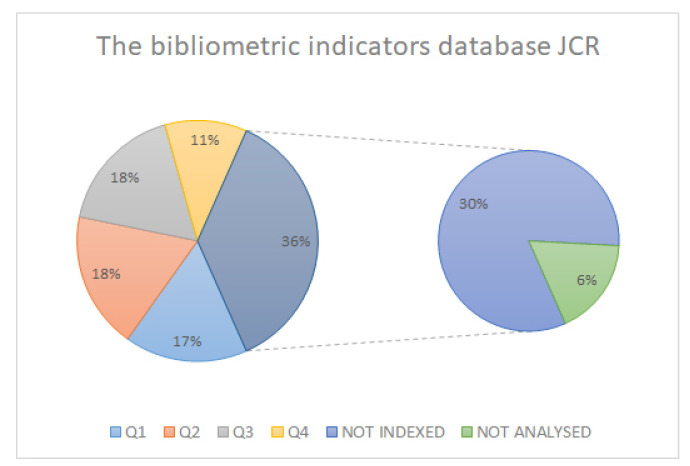
JCR database bibliometric indicators.

**Figure 4 sensors-21-06532-f004:**
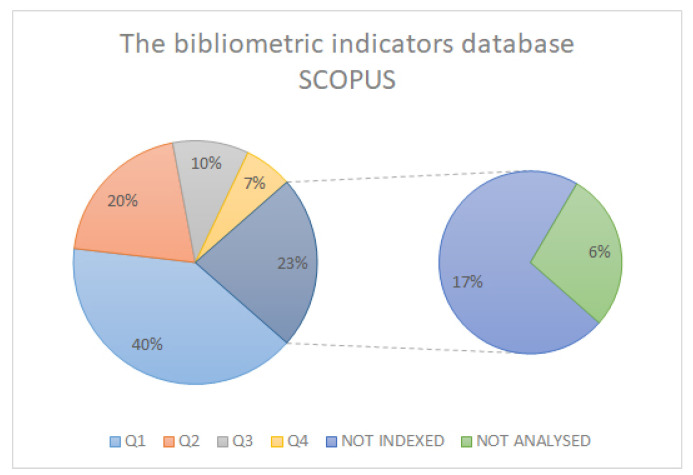
SCOPUS database bibliometric indicators.

**Figure 5 sensors-21-06532-f005:**
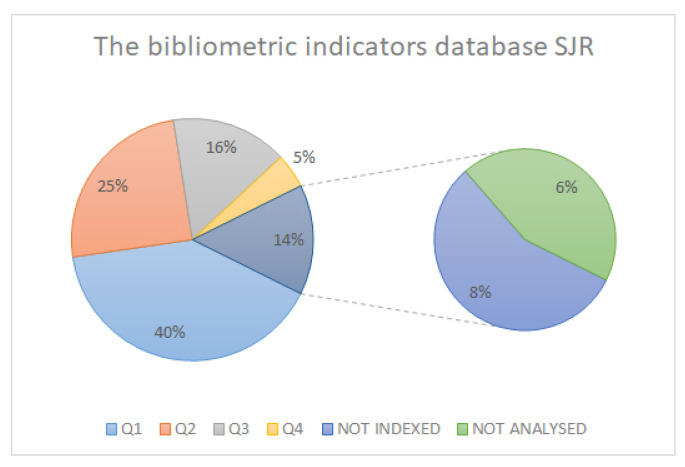
SJR database bibliometric indicators.

**Figure 6 sensors-21-06532-f006:**
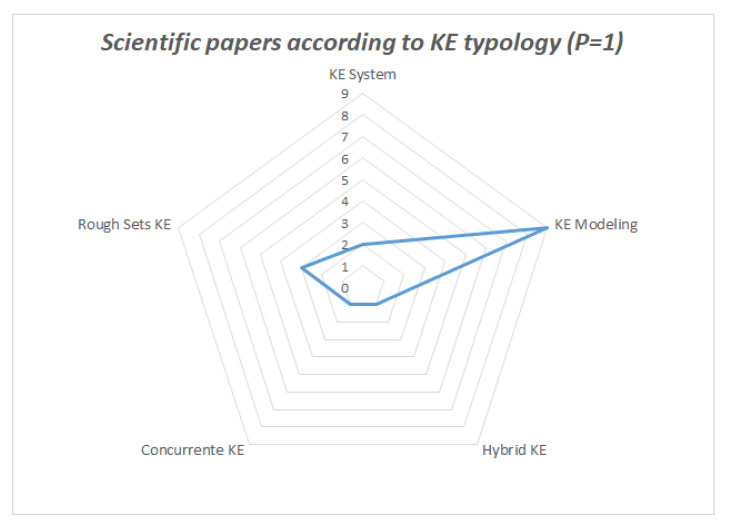
Incidence of KE typology on articles that obtained a score of P = 1.

**Figure 7 sensors-21-06532-f007:**
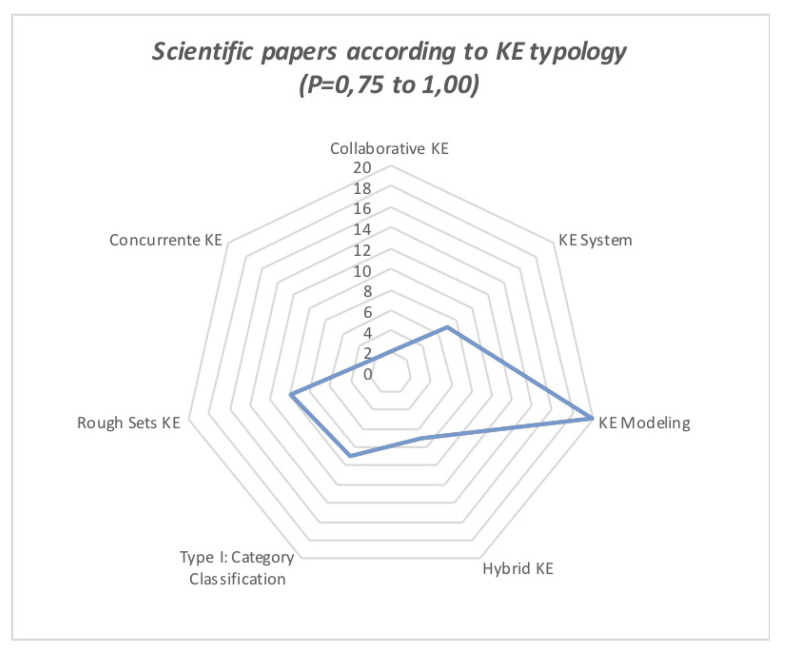
Incidence of KE typology on articles that obtained a score of P = 0.75 to P = 1.00.

**Figure 8 sensors-21-06532-f008:**
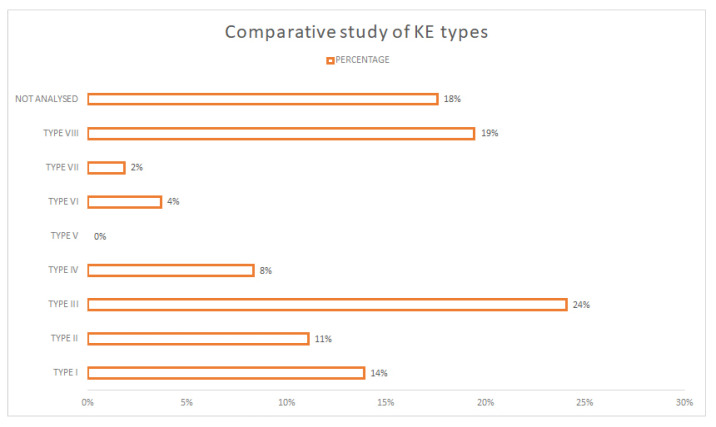
Comparative study of KE types.

**Figure 9 sensors-21-06532-f009:**
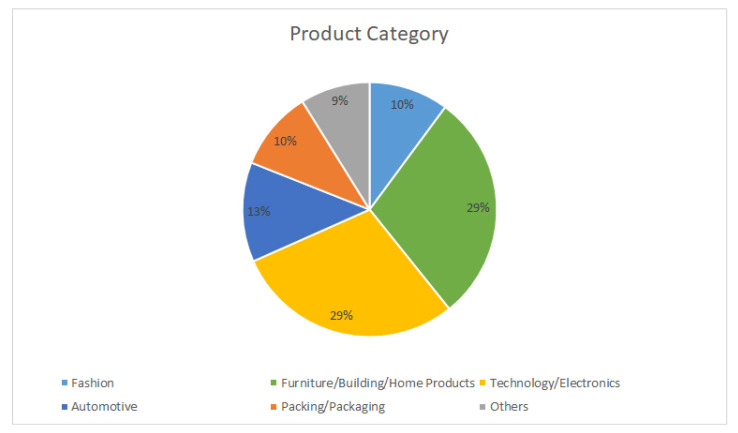
Classification by product category.

**Figure 10 sensors-21-06532-f010:**
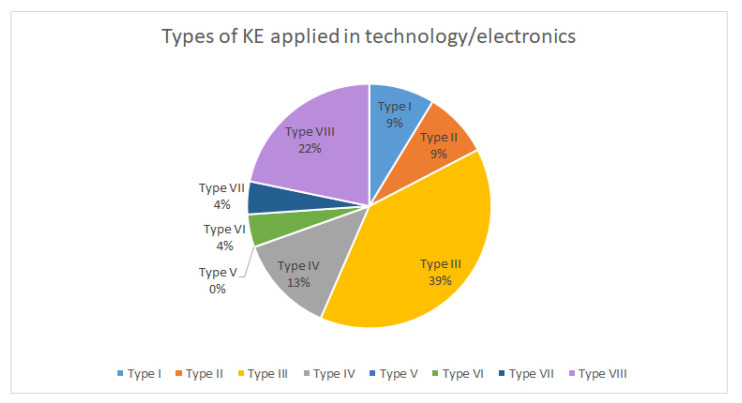
Classification by category of technological and electronic products.

**Figure 11 sensors-21-06532-f011:**
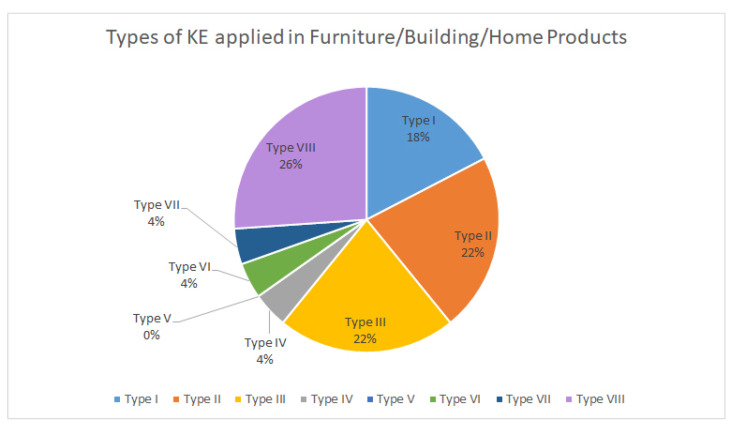
Classification by category of construction-related and household products.

**Figure 12 sensors-21-06532-f012:**
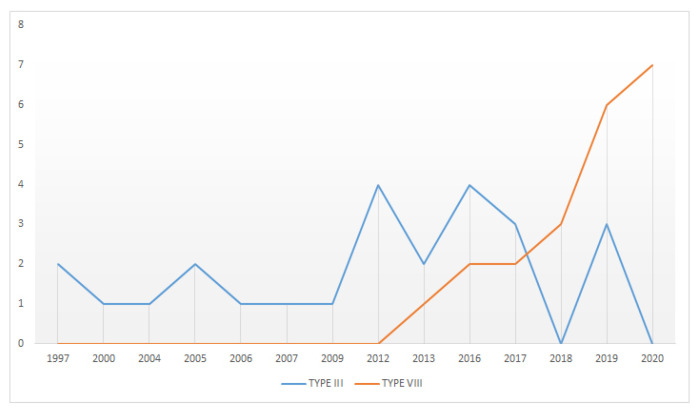
Comparative study of KE types.

**Table 1 sensors-21-06532-t001:** Selected journals and conferences.

Journal
Acta Microscopica
Advanced Engineering Informatics
Advanced Science Letters
Advances in Mechanical Engineering
AIEDAM, Artificial Intelligence, Design, Analysis and Manufacturing
Applied Ergonomics
Applied Sciences
ARPN Journal of Engineering and Applied Sciences
Baltic Journal of Road and Bridge Engineering
Building and Environment
CIRP Journal of Manufacturing Science and Technology
CoDesign
Color Research and Application
Communications in Science and Technology
Computer Science
Computer-Aided Design and Applications
Computers and Industrial Engineering
Computers and Operations Research
Computers and Industrial Engineering
Computers in Industry
CONCURRENT ENGINEERING: Research and Applications
Displays
Engineering Applications of Artificial Intelligence
European J. Industrial Engineering
Expert System witn Apllications Food Quality and Preference
GSIM WORKING PAPERS
Human Factors and Ergonomics In Manufacturing
Indonesian Journal of Electrical Engineering and Computer Science
International Conference on Rough Sets and Current Trends in Computing
International Journal of Clothing Science and Technology
International Journal of Industrial Ergonomics
Journal of Advanced Mechanical Design System and Manufacturing
Journal of Engineering Design
Product Planning and Control
Research in Engineering Design

**Table 2 sensors-21-06532-t002:** Analysed articles and extracted data.

		JCR Data		Scopus Data		SJR Data	
Year	Journal	Quartile	I. Factor	Quartile	I. Factor	Quartile	I.Factor
2021	Research in Engineering Design	Q2	2.224	Q1	5.2	Q1	0.814
2020	Color Research and Application	Q4	1.091	Q3	2.2	Q2	0.369
…	…	…	…	…	…	…	…
2019	International Journal of Scientific and Technology Research	-	-	Q4	0.2	Q3	0.123
…	…	…	…	…	…	…	…
2018	Engineering Applications of Artificial Intelligence	Q1	4.201	Q1	8.0	Q1	1.011
…	…	…	…	…	…	…	…
2017	Research in Engineering Design	Q2	2.224	Q1	5.2	Q1	0.814
…	…	…	…	…	…	…	…
2016	Journal of Ambient Intelligence and Humanized Computing	Q3	1.588	Q1	3.0	Q1	0.544
…	…	…	…	…	…	…	…
2015	Journal of Interdisciplinary Mathematics	-	-	Q4	0.1	Q4	0.146
…	…	…	…	…	…	…	…
2014	Baltic Journal of Road and Bridge Engineering	Q3	0.766	Q3	1.6	Q3	0.374
…	…	…	…	…	…	…	…
2013	Research Journal of Applied Sciences, Engineering and Technology	-	-	-	-	Q3	0.147
…	…	…	…	…	…	…	…
2012	International Journal of Industrial Ergonomics	Q2	1.208	Q1	3.5	Q1	1.021
…	…	…	…	…	…	…	…
2011	Applied Ergonomics	Q1	1.728	Q1	3.8	Q1	1.197
2010	Computers and Industrial Engineering	Q1	1.543	Q1	4.1	Q1	1.210
2009	Journal of Engineering Design	Q1	1580	Q1	2.2	Q1	0.591
…	…	…	…	…	…	…	…
2005	Computers and Operations Research	Q2	0.746	-	-	Q1	1.157
…	…	…	…	…	…	…	…
1997	International Journal of Industrial Ergonomics	Q2	0.387	Q1	3.0	-	-

**Table 3 sensors-21-06532-t003:** Extracted data from selected papers.

ID	Title	Authors	Date	Journal	QAQ1 20%	QAQ2 30%	QAQ3 10%	QAQ4 30%	QAQ5 10%	TOTAL SCORE 100%
**S1**	A fuzzy mapping method for Kansei needs interpretation considering the individual Kansei variance	Dong et al.	2021	Research in Engineering Design	Y	Y	Y	N	N	0.60
**S2**	Research on product color design decision driven by brand image	Zhand et al.	2020	Color Research and Application	N	Y	Y	N	Y	0.50
**S3**	An exploratory study on computer-aided affective product design based on crowdsourcing	Chu et al.	2020	Journal of Ambient Intelligence and Humanized Computing	P	Y	Y	Y	P	0.85
**S4**	A user biology preference prediction model based on the perceptual evaluations of designers for biologically inspired design	Luo et al.	2020	Symmetry	Y	Y	N	N	N	0.50
**S5**	Subjective product evaluation system based on kansei engineering and analytic hierarchy process	Zuo and Wang	2020	Symmetry	N	Y	Y	Y	Y	0.80
**S6**	Ergonomic adaptability design of classroom microscope based on kansei engineering	Chen et al.	2020	Acta Microscopica	Y	N	Y	N	Y	0.40
**S7**	Research on the construction method of kansei image prediction model based on cognition of EEG and ET	Yang et al.	2020	International Journal on Interactive Design and Manufacturing	P	Y	Y	Y	Y	0.90
**S8**	An investigation into 2D and 3D shapes perception	Čok et al.	2020	Tehnicki Vjesnik	Y	Y	Y	N	P	0.65
**S9**	Research on color optimization of tricolor product considering color harmony and users’ emotion	Guo et al.	2020	Color Research and Application	Y	Y	Y	N	Y	0.70
**(…)**	**(…)**	**(…)**	**(…)**	**(…)**	**(…)**	**(…)**	**(…)**	**(…)**	**(…)**	**(…)**
**S105**	A semantic differential study of designers’ and users’ product form perception	Hsu et al.	2000	International Journal of Industrial Ergonomics	Y	Y	Y	P	P	0.80
**S106**	Kansei engineering research on the design of construction machinery	Nakada	1997	International Journal of Industrial Ergonomics	Y	Y	Y	Y	Y	1.00
**S107**	Application studies to car interior of Kansei engineering	Jindo and Hirasago	1997	International Journal of Industrial Ergonomics	Y	Y	Y	Y	Y	1.00
**S108**	A Kansei Engineering approach to a driver/vehicle system	Horiguchi and Suetomi	1995	International Journal of Industrial Ergonomics	-	-	-	-	-	-
**S109**	Development of a design support system for office chairs using 3-D graphics	Jindo et al.	1995	International Journal of Industrial Ergonomics	-	-	-	-	-	-

**Table 4 sensors-21-06532-t004:** Weightings of QAQs.

QAQ1	QAQ2	QAQ3	QAQ4	QAQ5
20%	30%	10%	30%	10%

**Table 5 sensors-21-06532-t005:** Number of items per score range.

		>0–0.25	>0.25–0.5	>0.5–0.75	>0.75–1	=100
No. of articles	109					
Not found	22					
Total papers analysed	87	3	12	14	**57**	16
		3%	14%	16%	**66%**	18%

**Table 6 sensors-21-06532-t006:** Incidence of KE typology on articles that obtained a score of P = 1.

TYPE OF KE (P = 1)	N°	%
KES (KE System)	2	13%
KE Modeling	9	56%
Hybrid KE	1	6%
Concurrente KE	1	6%
Rough Sets KE	3	19%

**Table 7 sensors-21-06532-t007:** Incidence of KE typology on articles that obtained a score of P = 0.75 to P = 1.00.

TYPE OF KE (P = 75–100)	N°	%
Collaborative KE	2	4%
KES (KE System)	7	12%
KE Modeling	**20**	**35%**
Hybrid KE	7	12%
Category Classification	9	16%
Rough Sets KE	10	18%
Concurrente KE	2	4%

## Data Availability

Not applicable.

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
