# Peer review of "Systematic Literature Reviews in Kansei Engineering for Product Design—A Comparative Study from 1995 to 2020†"

_sensors, 2021, doi:10.3390/s21196532_

Round 1
Reviewer 1 Report
This manuscript’s idea is very interesting and I recommend this manuscript to be published after some revision:
1.In Line 47 (Introduction),"It includes the eight methods used in each of the KE implementation phases". This sentence seems to be logically inconsistent with the Figure 1.
2.The Selection of studies in Section 3 (Methodology) is unclear. This Section use a lot of words to describe the filter results. I suggest authors reorganize these parts and add a research flowchart to describe the procedure.
3.In 3.5 Data extracted from the selected paper. Please explain the meaning of CC1, CC2,CC3,CC4,CC5,N,S,P in Table 3.
4. In Section 4 (Results and Discussion), Figure 5 and 6 seems to be logically inconsistent with Table 6 and 7.
5.In Section 5 (Conclusion), the second paragraph seems more like findings of the manuscript, but not the conclusions. Please supplement more summarize the research.
Reviewer 2 Report
The manuscript deals with an interesting topic where a systematic literature review is conducted on the most relevant methodologies based on Kansei Engineering, i.e. emotional design.
Title:
- I recommend changing the title to become more attractive and informative.
Abstract:
- I recommend avoiding using the term prove. Instead, please, use substantiate.
Introduction:
- The beginning of the introduction is very general. I recommend adding more relevant references where the authors would derive from the societal happening. Please, be more concrete. What happens on the market with this regard? Why this topic is worth researching?
- Why the authors explain so in detail Kansei Engineering in the introduction? I suggest moving the content to the Background section.
- If one study on review of emotional design methodologies already exists – what are the shortcomings, so that the authors decided for SLR?
Background Section:
- The beginning of this section is quite poor. There are many very short paragraphs. I recommend the authors rewrite it.
- In overall, the whole background section is very poor. The authors do not discuss the main concepts as necessary.
Methodology:
- Why did the authors focus on Scopus and WoS databases? Why not examining more databases where the duplicates would be eliminated?
- Please, add some diagram, e.g. Prisma flow diagram where it would be seen how you excluded the studies.
- How were the possible biases managed? Did only one researcher make a review of publications? Why not three of them?
- In what period, when exactly SLR was conducted?
- Why not the papers from 2022 were not included in SLR?
- How did the authors select keywords to form the search string? If effect was included, why not affect, influence, impact were included? Was there any optimisation process?
- The authors considered methodology in publications. I miss the statement what they consider for methodology. How did they "group” various approaches to specific methodological groups?
Results:
- I recommend more in-depth approach in presenting the results. What is the new scientific contribution?
- Please, elaborate the discussion and provide more in-depth conclusions.
Conclusion:
- The authors should reconsider limitations of SLR. How did the authors limit or not limit to specific field, e.g. smart devices?
- Who can benefit from the findings?
- Please, provide theoretical and practical limitations.
Reviewer 3 Report
Kansei engineering is a research area which have been developed for over 40 years. We can observe that the research topics and methodologies are already encountered barriers and there is a urging need to seek for breaking-through.
Although this study has carried out examinations carefully on recent literatures, the criteria for selecting KE studies is quite simple and naive.
The authors can refer to the study "A Systematic literature review of consumers' cognitive-affective needs in product design from 1999 to 2019" for comparison.
Round 2
Reviewer 2 Report
I commend the authors for improvements of the manuscript. I recommend the paper for publication.
Author Response
We want to thank the reviewer for their work.